# The PM6-FGC Method: Improved Corrections for Amines and Amides

**DOI:** 10.3390/molecules27051678

**Published:** 2022-03-03

**Authors:** Martiño Ríos-García, Berta Fernández, Jesús Rodríguez-Otero, Enrique M. Cabaleiro-Lago, Saulo A. Vázquez

**Affiliations:** Departamento de Química Física, Universidade de Santiago de Compostela, 15782 Santiago de Compostela, Spain; martino.rios@rai.usc.es (M.R.-G.); berta.fernandez@usc.es (B.F.); r.otero@usc.es (J.R.-O.); caba.lago@usc.es (E.M.C.-L.)

**Keywords:** semiempirical methods, PM6 Hamiltonian, PM6-FGC approach, noncovalent interactions, potential energy curves

## Abstract

Recently, we reported a new approach to develop pairwise analytical corrections to improve the description of noncovalent interactions, by approximate methods of electronic structures, such as semiempirical quantum mechanical (SQM) methods. In particular, and as a proof of concept, we used the PM6 Hamiltonian and we named the method PM6-FGC, where the FGC acronym, corresponding to Functional Group Corrections, emphasizes the idea that the corrections work for specific functional groups rather than for individual atom pairs. The analytical corrections were derived from fits to B3LYP-D3/def2-TZVP (reference). PM6 interaction energy differences, evaluated for a reduced set of small bimolecular complexes, were chosen as representatives of saturated hydrocarbons, carboxylic, amine and, tentatively, amide functional groups. For the validation, the method was applied to several complexes of well-known databases, as well as to complexes of diglycine and dialanine, assuming the transferability of amine group corrections to amide groups. The PM6-FGC method showed great potential but revealed significant inaccuracies for the description of some interactions involving the –NH_2_ group in amines and amides, caused by the inadequate selection of the model compound used to represent these functional groups (an NH_3_ molecule). In this work, methylamine and acetamide are used as representatives of amine and amide groups, respectively. This new selection leads to significant improvements in the calculation of noncovalent interactions in the validation set.

## 1. Introduction

The development of corrections for the description of noncovalent interactions by semiempirical quantum mechanical (SQM) methods has attracted much attention in the last two decades. It is worth mentioning the work carried out by Hobza et al. [1,2,3,4], through the development of several generations of corrections for dispersion [1,2,4], hydrogen bond [2,4] and halogen bond [3] interactions, and their parameterization within the PM6 method [5]. Korth [6] and Jensen and co-workers [7] contributed to the third generation of this series.

Truhlar, Gao, and their co-workers, implemented the polarized molecular orbital (PMO) method [8,9,10,11], based on a Neglect of Diatomic Differential Overlap (NDDO) Hamiltonian, which includes polarization functions on hydrogen atoms. Additionally, they improved the description of dispersion interactions by adding the first damped dispersion term developed by Grimme [12,13]. The final versions of the PMO method, namely, PMO2 [10] and PMO2a [11], may accurately describe polarization effects, as well as noncovalent complexation energies for some specific systems. 

Significant contributions were also made by Thiel et al., who improved the semiempirical Hamiltonian through the ortogonalization-corrected methods OMx [14,15,16] and ODMx [17], which led to better results in comparison to NDDO-based methods (e.g., AM1 [18] or PM6 [5]). These semiempirical Hamiltonians needed to incorporate dispersion corrections to improve the description of noncovalent interactions. For the recent ODMx methods, several training sets were considered, including S66 dataset [19,20], and the followed procedure is extensively discussed in the work by Dral et al. [17]. 

In general, the corrections included in the above studies for dispersion and hydrogen bonding interactions are modelled on potential functions, based on physically sound formulas and parameterized using quite large training sets, which when properly selected, can ensure wider applicability range. The above studies considerably improved the evaluation of noncovalent interactions using SQM methodology, but they still show significant deficiencies in some cases [21]. Often these errors depend on the relative orientation of the interacting molecules and can be attributed to possible shortcomings in popular data sets, which in general only include the most relevant interacting configurations of the molecules. 

Very recently, we developed the PM6 Functional Group Corrections (PM6-FGC) approach [21] to correct the semiempirical PM6 Hamiltonian and improve the description of noncovalent interactions. Although in general, the subsequent PM7 method [22] provides higher accuracy than its predecessor, we used the PM6 Hamiltonian, following Grimme and co-workers’ recommendation [23]. Specifically, they reported significant errors in interaction energies calculated with PM7 for large systems, due to the damping and truncation of the dispersion energy at long range in this Hamiltonian [24,25]. Our previous work [21], presented as a proof-of-concept study, provides an alternative way to obtain analytical corrections for SQM methods. The improvement of the description of noncovalent interactions in SQM methods is particularly important in quantum mechanics/molecular mechanics (QM/MM) simulations, since the efficiency of the quantum mechanical method is crucial for practical applications. Furthermore, the use of a semiempirical Hamiltonian may be convenient for investigating chemical reactions because the semiempirical parameters may be specifically adjusted for individual reactions [26].

In our previous work [21], the corrections are obtained by fits involving intermolecular potential energy curves (IPECs) of bimolecular complexes of small molecules that contain functional groups of relevance for biological chemistry. Specifically, we considered methane, formic acid, and ammonia as representatives of saturated hydrocarbons, carboxylic acids, and –NH_2_ groups in amines and, tentatively, in amides. We evaluated IPECs for the six different pair combinations of these molecules, considering orientations of the interacting molecules that emphasize the atom–pair interactions in the complexes. 

The reference data for the fittings were obtained by calculations using Density Functional Theory (DFT), with the B3LYP functional [27,28,29], including the D3 dispersion correction with the Becke–Johnson damping scheme [30,31,32], and the valence triple-zeta polarization def2-TZVP basis set [33]. This combination of density functional and basis set was selected after benchmarking the results with those obtained at the CCSD(T) level [34], together with the augmented correlation consistent polarized valence triple-zeta basis set aug-cc-pVTZ [35]. The differences between the DFT and the coupled cluster IPECs were negligible for our purposes and showed the efficiency of the combination of the B3LYP-D3 density functional and the def2-TZVP basis set, in providing acceptable and inexpensive interaction energies. Since we plan to extend the study to larger molecules and other types of functional groups, the use of an efficient and inexpensive reference methodology is important and, therefore, we have selected the B3LYP-D3/def2-TZVP level as the reference.

Our analytical pairwise corrections are parameterized by fits to a series of curves obtained by the difference between the IPECs calculated at the B3LYP-D3/def2-TZVP level and those computed with the PM6 method, following a strategy similar to that employed in the derivation of intermolecular potentials for dynamics studies [36,37,38,39,40]. Our previous work [21] emphasized the importance of including all the significant orientations of the interacting molecules in the databases, a fact that is crucial in the development of well-balanced corrections. Furthermore, the use of general corrections is important to take into account that SQM methods have significant limitations, not only to accurately describe dispersion interactions, but also other contributions, such as electrostatics, induction, and exchange repulsion. 

The IPECs obtained with the PM6-FGC method for the bimolecular complexes of methane, formic acid, and ammonia acceptably reproduced those determined at the DFT level [21], and considerably improved the results provided by previously corrected SQM methods (particularly, PMO2 and PM6-D3H4). To assess the performance of the method, as well as the transferability of the corrections to other systems, we carried out the PM6-FGC evaluation of interaction energies for several complexes of the S66 [19], A24 [41], and ADIM6 [42,43] data sets, as well as for a collection of different conformers of the diglycine dimer and trimer, and the dialanine dimer, obtained through automated exploration of the corresponding potential energy surfaces (PESs). Although the performance of the PM6-FGC approach was acceptable in general, we found significant inaccuracies in the description of some interactions involving the –NH_2_ group in amines and amides (see [21] for extra details). The aim of the present study is to improve the method by handling and solving these shortcomings.

The manuscript is organized as follows: in Section 2, the results are presented and discussed, in Section 3, we describe the method and give the computational details, and in the last Section, we summarize and conclude.

## 2. Results and Discussion

In this work, for the parameterization of our corrections, we have selected the following model compounds: acetamide, methylamine, methane, formic acid and acetic acid. We considered the different atom types defined in Figure 1a. For parameterizations involving the carboxylic functional group, we used either formic acid, as in our previous work [21], or acetic acid. The latter compound was selected to explore different alternatives in our parameterization strategy, as explained later. As discussed in ref. [21], our main purpose is to develop corrections to improve the description of noncovalent interactions in peptides and other biological systems. As for the development of molecular mechanics force fields, we assume that the corrections developed from fits involving small model compounds may be transferable to larger systems containing the corresponding functional groups. Figure 1b depicts a conformation of dialanine showing the atom types defined in this study. 

We have performed fits for the following complexes: CH_3_NH_2_–CH_3_NH_2_, CH_3_CONH_2_–CH_3_CONH_2_, CH_3_NH_2_–CH_3_CONH_2_, CH_3_NH_2_–CH_4_, CH_3_NH_2_–HCOOH, CH_3_CONH_2_–CH_4_, and CH_3_CONH_2_–CH_3_COOH. For each complex, we carried out a series of independent fits, adjusting all the parameters simultaneously. The final parameters, resulting from our best fits, are given in the Appendix A, in the form of a Python dictionary. Notice that, for each complex, the total number of parameters involved in the fits is five times the number of different types of pairwise interactions. 

### 2.1. Methylamine Dimer

The methylamine molecule has four different types of atoms; therefore, the dimer exhibits ten different types of pairwise interactions. For the fits, we considered ten orientations that emphasize the different types of two-body interactions, as well as a cyclic configuration that facilitates the simultaneous interaction between most of the atoms (orientation 11). Four of these orientations are depicted in Figure 2, together with the corresponding intermolecular potential energy curves. The remaining five orientations and the associated IPECs are shown in Appendix A. In these and the following figures, the B3LYP-D3/def2-TZVP data are displayed as open, black circles, and *r* refers to the distance between the selected pair of attacking atoms. The IPECs calculated by the PM6 method are shown as red solid lines. For comparison, we include the results obtained with the PM6-D3H4 method (blue lines), which is one of the most successfully applied SQM methods. For this system, 50 independent parameters were fitted simultaneously. The parameters of the best fit are collected in the Appendix A, in the form of a Python dictionary. Adding the corrections to the PM6 Hamiltonian results in the PM6-FGC method, the corresponding IPECs are shown as green lines in the figures.

In orientation 2, an amine hydrogen of one of the molecules interacts with the nitrogen of the other molecule in a way that favors the formation of a hydrogen bond. As expected, this orientation leads to the deepest well among those considered in the present study. The B3LYP-D3 interaction energy is −15.4 kJ/mol at 2.3 Å (distance between the attacking atoms, i.e., H and N). Both corrected methods, that is, PM6-D3H4 and PM6-FGC, reproduce the DFT IPEC quite well. The PM6 method underestimates the strength of the hydrogen bond, which is one of the well-known deficiencies of this method. The PM6-D3H4 method clearly overestimates the strength of the interaction in the well region for orientations 3, 4, and 5. Hobza and co-workers used the S66 database [19,20] for the parameterization of the D3H4 corrections. In general, the interaction energies included in this database were obtained by CCSD(T) calculations, with complete basis set extrapolation and, therefore, this should be borne in mind when the PM6-D3H4 IPECs are compared with our DFT data. However, as previously shown [21], for the type of model compounds considered in our work, the results of B3LYP-D3/def2-TZVP calculations are, in general, in very good agreement with those determined at the CCSD(T)/CBS level (see also Section 2.8). 

The case of orientation 5 is particularly interesting. This orientation stresses the interaction between amine hydrogens, which is repulsive at the B3LYP-D3 level. The PM6-D3H4 method predicts a significant well depth, similar to that found for the ammonia dimer [21]. Our method also gives a minimum, but the well depth is rather small (1.8 kJ/mol). Regarding the methylamine dimer, this is perhaps the orientation for which our method shows the most significant deviation from the reference data. 

In general, for the 11 orientations considered here, the performance of our method is quite good and the improvement over the PM6 and PM6-D3H4 is clear. This may be quantified by the values of the mean absolute errors (MAEs), which are collected in Appendix A. For interaction energies below 5 kJ/mol, the calculated MAEs are 1.65 kJ/mol (PM6), 1.32 kJ/mol (PM6-D3H4), and 0.34 kJ/mol (PM6-FGC). Increasing the maximum interaction energy up to 20 kJ/mol, we obtained 2.69 kJ/mol (PM6), 2.27 kJ/mol (PM6-D3H4), and 0.38 kJ/mol (PM6-FGC). The significant increase in the PM6 and PM6-D3H4 MAEs, obtained by increasing the range of the interaction energies, indicates that the accuracy of these methods for the repulsive region of the intermolecular potential is very low, as could be anticipated by close inspection of Figure 2 and Appendix A. The performance of the PM6-FGC is quite good, even up to interaction energies around 100 kJ/mol; at this energy, the calculated MAEs are 4.79 kJ/mol (PM6), 5.15 kJ/mol (PM6-D3H4), and 0.41 kJ/mol (PM6-FGC). For higher repulsive energies, the PM6-FGC may fail dramatically with the present corrections, although the development of specific corrections for high interaction energies is straightforward. For most practical applications, however, the attractive region of the potential energy is the important one.

### 2.2. Acetamide Dimer

For acetamide, we defined five different atom types, as shown in Figure 1. Consequently, the dimer has 21 different types of pairwise interactions. For the fits, we considered 22 orientations. The additional one (i.e., orientation 22) corresponds to the most stable conformation of the dimer. For this orientation, we performed a full optimization at the B3LYP-D3/def2-TZVP level. Then, we calculated the associated IPEC by varying one of the O⋯H distances along the direction defined by these two atoms. In Figure 3, we display eight selected orientations with the corresponding IPECs. The IPECs of the remaining orientations are shown in Appendix A. For the most stable configuration of the dimer, both corrected methods reproduce the reference IPEC very well. The interaction energy at the minimum is about −70 kJ/mol. The underestimation of the hydrogen bonding interaction by the PM6 method is apparent in the figure. 

For some orientations (e.g., 4, 10, or 13), the PM6-D3H4 method overestimates the well depth of the minima. In addition, and resembling the results obtained for the methylamine dimer, the PM6-D3H4 method predicts a significant minimum for the orientation in which amide hydrogens face each other (orientation 19; cf. orientation 5 in Figure 2). The MAEs calculated for this complex for interaction energies below 5 kJ/mol are 2.51 kJ/mol (PM6), 1.55 kJ/mol (PM6-D3H4), and 0.54 kJ/mol (PM6-FGC). Again, the MAEs increase as we increase the upper limit of the interaction energy (see Appendix A). For the PM6-FGC method, the increase in the MAE is small; specifically, for interaction energies bellow 40 kJ/mol, the calculated MAE is 0.76 kJ/mol. In general, the performance of the PM6-FGC is quite good. However, for orientation 13, in which the oxygen of one of the molecules faces the nitrogen atom of the other molecule, our method significantly overestimates the interaction in the attractive region.

### 2.3. Methylamine—Acetamide Complex

The number of orientations considered for the CH_3_NH_2_–CH_3_CONH_2_ complex is 24, which is equal to the number of different types of pairwise interactions in this system. We show in Figure 4 the IPECs of eight orientations; the remaining IPECs are depicted in Appendix A. Among the orientations considered for this complex, orientation 13 exhibits the strongest attractive interaction, with a well depth of 28.9 kJ/mol at the DFT level, which is consistent with formation of a hydrogen bond between an amide hydrogen and the amine nitrogen of the partner molecule. For this orientation, both corrected methods reproduce the reference IPEC quite well. By contrast, and as expected, the PM6 method significantly underestimates the strength of the hydrogen bonding interaction.

For several orientations, the PM6 and PM6-D3H4 IPECs deviate markedly from the reference curves. Regarding the PM6-D3H4 method, most of the deviations involve overestimation of minimum well depths (see Figure 4), which is the opposite of the trend shown by the PM6 method (see Appendix A). For orientation 14, in which an amine hydrogen faces an amide hydrogen, the PM6-D3H4 method predicts a significant minimum, which contrasts with the repulsive nature of this orientation, as predicted by the B3LYP-D3/def2-TZVP calculations. This result resembles those found for orientations 5 and 19 in the methylamine and acetamide dimers, respectively. The most significant deviations observed for our method correspond to orientations 2, 23 (Figure 4), 8 (Appendix A), and 19 (Appendix A), but they are rather small. These results can be quantified by the MAEs (Appendix A). For interaction energies below 5 kJ/mol, the calculated MAEs are 1.80 kJ/mol (PM6), 1.71 kJ/mol (PM6-D3H4), and 0.52 kJ/mol (PM6-FGC).

### 2.4. Methylamine—Methane Complex

Since methylamine and methane have four and two different atom types, respectively, the number of different types of pairwise interactions in the CH_3_NH_2_–CH_4_ complex is eight. This is the number of orientations we considered for this complex. Figure 5 shows the IPECs for four selected orientations. The IPECs for the remaining orientations are depicted in Appendix A of the Appendix A. As can be seen, although the PM6-D3H4 curves for orientations 1 and 5 show small disagreement with the reference IPECs, both corrected methods describe the intermolecular interaction satisfactorily. The PM6 curves deviate significantly from the reference DFT data, especially in orientations 4 and 8, in which hydrogen atoms face each other.

The MAEs calculated for interaction energies below 5 kJ/mol are 0.63 kJ/mol (PM6), 0.47 kJ/mol (PM6-D3H4), and 0.18 kJ/mol (PM6-FGC). If we increase the energy range for the calculation of MAEs, the values for the PM6 and PM6-D3H4 methods increase significantly (see Appendix A), thus, following the same trend as that found for the previous complexes.

### 2.5. Methylamine—Formic Acid Complex

For this complex, we considered 20 different orientations, which is the number of different types of pairwise interactions exhibited in this system. In Figure 6, we display the IPECs for six orientations of this complex. The remaining 14 orientations are depicted in Appendix A. Orientation 4 shows the strongest attractive interaction, as expected, since it leads to hydrogen bonding between the carboxylic hydrogen and the amine nitrogen. The well depth calculated at the B3LYP-D3/def2-TZVP level is 43.5 kJ/mol, with the minimum being at an H⋯N distance of 1.8 Å. In general, the performance of the PM6-FGC is quite good; the most significant deviations from the reference curves, for example, those occurring for orientations 6 and 19, are rather small.

For several orientations, the PM6-D3H4 curves show remarkable discrepancies with the reference IPECs. This is the case for orientations 1 and 6, in which the carboxylic carbon faces the nitrogen atom and the amine hydrogen, respectively. Resembling the results found in previous complexes, wherein hydrogen bonding can take place, the most striking result appears for orientation 9, in which the carboxylic hydrogen faces one of the amine hydrogens. This orientation is repulsive at the B3LYP-D3/def2-TZVP level, but the PM6-D3H4 method predicts a clear minimum (4.8 kJ/mol at an H⋯H distance of 1.9 Å). We found the same behaviour for the NH_3_–HCOOH complex in our previous work [21]. As for the complexes discussed earlier in this paper, the improvement of our approach over the PM6 and PM6-D3H4 methods is clear. Considering all the orientations of this complex, the MAEs calculated for interaction energies below 5 kJ/mol are 1.76 kJ/mol (PM6), 1.88 kJ/mol (PM6-D3H4), and 0.38 kJ/mol (PM6-FGC). For this system, the PM6 value is slightly lower than that calculated for the PM6-D3H4 method, and the same occurs for the other upper limits of the interaction energies (Appendix A).

### 2.6. Acetamide—Methane Complex

This complex shows 12 different types of pairwise interactions. Accordingly, we considered 12 orientations, selected in such a way that they emphasize the different pair interactions. The IPECs for six of these orientations are exhibited in Figure 7; the IPECs of the remaining orientations are shown in Appendix A. As for the CH_3_NH_2_–CH_4_ complex, both corrected methods afford results in good or reasonably good agreement with the reference IPECs. The improvement of PM6-FGC over the PM6-D3H4 method is reflected in the values of the calculated MAEs (Appendix A). Considering all the orientations and interaction energies below 5 kJ/mol, the MAE of our method (0.15 kJ/mol) is four times smaller than that of the PM6-D3H4 method (0.61 kJ/mol).

### 2.7. Acetamide—Acetic Acid Complex

For this complex, we explored three different parameterization strategies. In one of them, the methyl group of the acetic acid molecule was considered to be equivalent to that of acetamide (see Figure 1a), and the parameters for the CA–CA, CA–HCA, and HCA–HCA interactions (15 in all) were taken from the parameterization of the acetamide dimer. With this definition of atom types, there are 35 different types of pairwise interactions and a total of 175 parameters, so that the fittings involved the simultaneous adjustment of 160 parameters. In the second one, in addition to the methyl group, the carbon and oxygen atoms of the carbonyl group of the acetic acid molecule were considered to be equivalent to the corresponding atoms in acetamide. This leads to 30 different types of pairwise interactions and a total of 150 parameters. Half of the distinct types of pairwise interactions are exhibited in the acetamide dimer, and the corresponding 75 parameters were taken from that system. Consequently, only 75 parameters were fitted in this case. Finally, in the third strategy, all the atom types in acetic acid were different from those of acetamide, which resulted in 36 distinct types of pairwise interactions and 180 parameters in all. Preliminary fits showed that the first and the third strategies led to similar values of the objective function see the section on Methods and Computational Details), whereas the second strategy resulted in larger values. This may suggest that the carbonyl group in the acetic acid molecule behaves somewhat different from that in acetamide. For this reason, we chose the first strategy.

Even though for this system we followed the first parameterization strategy described in the previous paragraph, we considered 37 different orientations, 36 of which emphasize the distinct types of pairwise interactions in the complex, assuming no transferability of atom types between the two molecules (i.e., the assumption followed in the third parameterization strategy). The remaining orientation corresponds to the most stable configuration of the complex, which facilitates the formation of a double hydrogen bond (orientation 37). Figure 8 displays the IPECs of eight selected orientations; the results for the remaining orientations are shown in Appendix A. For many orientations, the failure of the PM6 and PM6-D3H4 methods is apparent. As for the previous complexes in which electrostatic interactions are important, for many orientations, the PM6 method underestimates the strength of attractive interactions in the well region (see Figure 8, Appendix A). 

The PM6-D3H4 method significantly overestimates the well depth for several orientations. This is the case for orientation 11, in which the carboxyl hydrogen attacks the carbon atom of the amide group. For this orientation, the PM6-D3H4 predicts a well depth 18.5 kJ/mol, whereas the DFT value is 2.8 kJ/mol. The PM6-FGC curve, with a well depth of 6.5 kJ/mol, also deviates significantly from the reference IPEC, although not as much as the PM6-D3H4 curve. Another significant failure of our method occurs for orientation 29, in which the carboxyl hydrogen faces one of the amide hydrogens. Although the PM6-FGC curve is repulsive, it is less repulsive than that predicted by the B3LYP-D3/def2-TZVP calculations. For this orientation, the PM6-D3H4 method shows a clear minimum, thus, resembling the behaviour found for other complexes (e.g., orientation 9 in the CH_3_NH_2_–HCOOH complex, Figure 6). For orientation 35, wherein the carboxyl hydrogen faces a methyl hydrogen of the partner molecule, the PM6-FGC curve around an H⋯H distance of 2 Å deviates somewhat from the reference curve. This deviation results from the strange behaviour of the PM6 Hamiltonian, whose curve shows a maximum at an H⋯H distance of 2.1 Å and a minimum at 1.5 Å. Nevertheless, in general, the performance of our method is reasonably good, and the improvement over the other SQM approaches is clear. The calculated MAEs, for energies below 5 kJ/mol, are 2.71, 1.72, and 0.51 kJ/mol for PM6, PM6-D3H4, and PM6-FGC, respectively (see Appendix A).

### 2.8. Validation of the PM6-FGC Method

One of the issues that deserves special attention in model fitting is the possibility of overfitting [44]. As discussed in our previous work [21], our tests suggested that our parameterization model does not show overfitting, at least at a significant level. Since this issue was analysed in detail in our previous work, here, we will only focus on the validation of the new corrections, by applying them to several complexes of the S66 database [19] that involve amine and amide groups, for which our previous corrections were found to be inaccurate, as well as to complexes of diglycine and dialanine, for which our method showed substantial improvement over the PM6 and the PM6-D3H4 methods.

Table 1 displays the interaction energies calculated for several complexes included in the S66 database, which contain amine and amide groups. The structures of these complexes are depicted in Appendix A. Notice that, in the S66 database, “peptide” refers to *N*-methylacetamide. In addition to the CCSD(T)/CBS benchmark values reported in ref. [19], we include the interaction energies calculated at the B3LYP-D3/def2-TZVP level, together with the values obtained with the PM6, PM6-D3H4, and PM6-FGC methods. For the PM6-FGC calculations, the atom types considered for the carbon and hydrogen atoms of the methyl group, attached to the nitrogen atom of *qu*-methylacetamide, were “CAN” and “HCAN”, respectively (see Figure 1). For our method, we also show the values obtained in our previous work. The first important observation is the very good agreement between the CCSD(T)/CBS data and the B3LYP-D3/def2-TZVP values, as previously mentioned. The corresponding MAE is below 1 kJ/mol. A further remarkable finding is the small MAE calculated for the PM6-D3H4 data. This is somewhat expected, since these structures were included in the training set used to parameterize this method. An analysis of the PM6-FGC results reveals a clear, general improvement of the corrections. The MAE calculated with the old corrections is more than twice the value obtained with the present corrections. Notice that in our previous work, we used NH_3_ as the representative for both amine and amide groups, which resulted in a drastic approximation for amides. Particularly, this led to a remarkable error for the global minimum of the acetamide dimer. It is important to mention that, in this work, we have included the most stable configuration of the acetamide dimer in the training set, but not the orientation of the methylamine dimer considered in the S66 database (complex 10, see Appendix A). Finally, the interaction energies calculated for complexes 46 and 62 with our previous corrections were found to be in better agreement with the benchmark data than those evaluated with the present corrections. This is probably a result of error compensation.

In the development of analytical corrections for biological compounds, we first focus on peptides. For this reason, for the validation of the PM6-FGC corrections, in this and the previous study, we have selected complexes of small peptides, specifically, the dimers of diglycine and dialanine, as well as the trimer of diglycine. The conformational space of these complexes was explored in our previous work, using AutoMeKin [45,46,47,48,49], software designed to predict and simulate the kinetics of complex reaction mechanisms. This program contains a module that allows one to locate minima for intermolecular complexes in an automated fashion. The search involves both intramolecular and intermolecular degrees of freedom, and for our study, we used the PM6-D3H4 method, available through the interface between AutoMeKin and the MOPAC2016 program [50].

As described in our previous study [21], for the dialanine dimer we obtained 90 different conformers. For these conformers, the interaction energies were evaluated by single-point B3LYP-D3/def2-TZVP calculations, using counterpoise corrections, and the data were used to compute linear correlations with the corresponding SQM values. The results are shown graphically in Figure 9. The black straight lines correspond to the case of perfect correlation. In addition, in seen in Table 2, we collected the calculated MAEs, as well as the mean bias errors (MBEs). The MBE is just the mean value of the differences between the reference and the model prediction (i.e., the SQM results). A positive MBE value indicates that the SQM method overestimates the strength of the interaction whereas a negative value indicates underestimation. As can be seen from the figure and the statistical values given in Table 2, our approach provides an improvement in the accuracy of SQM methods for the calculation of noncovalent interactions in this system. The MAE and MBE values calculated for PM6 and PM6-D3H4 are much larger than those computed for our method, using either the present or the previous corrections. The MAE calculated for the data obtained with the present corrections (4.9 kJ/mol) is significantly smaller than that determined in our previous study (8.7 kJ/mol). To a large extent, this is a consequence of the very small bias displayed by the interaction energies calculated with the present corrections.

The set of conformers of the diglycine dimer contain 77 different structures [21]. The linear correlations are depicted in Figure 10. The results for this system follow the same trends observed for the dialanine dimer. The PM6 method underestimates the strength of the noncovalent interactions (MBE = −15.0 kJ/mol) and, on the contrary, the PM6-D3H4 method substantially overestimates the interaction strength (MBE = 15.5 kJ/mol). Both methods give MAEs around 18 kJ/mol. Our method improves the statistics remarkably. With the present corrections, the calculated MAE is 4.0 kJ/mol, which is 33% smaller than our previous value. Furthermore, the MBE value is almost negligible (−0.6 kJ/mol), thus, pointing out a significant improvement over the previous results.

The last system considered here for the validation of our method is the trimer of diglycine. In this case, the interaction energy is calculated as the difference between the energy of the non-interacting monomer and dimer and the energy of the trimer. In our previous study, we found 146 different conformers for the trimer [21]. The linear correlations are depicted in Figure 11. For this system, the results obtained with the present corrections show a small bias (MBE = 4.6 kJ/mol) towards higher strength of the noncovalent interaction. However, the improvement over the previous parameterization is clear; the MAE and MBE values are almost half those determined in our previous work. In the analysis of these correlations, it should be kept in mind that the PM6-D3H4 method was parameterized using CCSD(T)/CBS data in general. Nonetheless, for these types of peptides, the higher accuracy of the PM6-FGC method is apparent.

## 3. Methods and Computational Details 

We followed the procedure described in our previous work [21], where we presented the PM6-FGC method, whose aim was to show a strategy to develop corrections able to satisfactorily model noncovalent interactions for all orientations of interacting molecules. In that work [21], to parameterize corrections involving N atoms, we selected the NH_3_ molecule as representative. We chose three molecules in all, specifically, methane, formic acid, and ammonia, which resulted in six bimolecular complexes needed to get the parameters. The corrections obtained from NH_3_/CH_4_ fittings could model –NH_2_/alkane interactions, but the use of ammonia as representative proved not to be adequate enough to reproduce other interactions of the –NH_2_ group in amines and amides (see for instance the results shown in Table 1 for some complexes of methylamine, *N*-methylacetamide and acetamide). Therefore, to try to solve these limitations, in the present investigation, we decided to use methylamine and acetamide as representatives, instead of ammonia. In addition, and with the purpose of exploring parameter transferability, we also considered the acetic acid molecule.

Since in ref. [21], we selected B3LYP-D3/def2-TZVP intermolecular potential energy curves for the training set, due to their good agreement with the corresponding CCSD(T)/aug-cc-pVTZ results, here, we used the same level of electronic structure for the evaluation of interaction energies in the new bimolecular complexes. As before, we employed the supermolecular approach and corrected for basis set superposition error (BSSE) with the counterpoise method [51,52]. The intramolecular geometries were obtained from B3LYP-D3/def2-TZVP optimizations and kept frozen in the interaction energy calculations. As shown in our previous work [21], the number and types of orientations play a crucial role in developing adequate well-balanced corrections. In this way, for each pair of molecules, the number of orientations should be at least equal to the number of the different pair–type interactions. The electronic structure calculations were performed with the ORCA 4.0 and 5.0 programs using the frozen core approximation [53,54].

The general expression for the noncovalent potential energy correction to the PM6 Hamiltonian is given as a pairwise sum of the form:(1)$$E_{\mathrm{corr}}=\underset{i}{\sum }\underset{j}{\sum }f_{\mathrm{cut}}\left(r_{ij}\right)\times \left\{A_{ij}e^{-B_{ij}r_{ij}}+\frac{C_{ij}}{r_{ij}^{D_{ij}}}\right\}$$
where indexes *i* and *j* refer to atoms belonging to different interacting molecules, and *r_ij_* is the interatomic distance between atoms *i* and *j*. The parameters *A_ij_*, *B_ij_*, *C_ij_* and *D_ij_* depend on the nature of the considered pair of atoms. The parameters *A_ij_* and *C_ij_* may be either positive or negative. In our previous work [21], we constrained the *D_ij_* parameters to be integers. In this study, to get additional, although small, flexibility, these parameters were considered real (positive) numbers instead. $f_{\mathrm{cut}}\left(r_{ij}\right)$ is a cutoff function introduced to remove the correction at very short *r_ij_* distances: (2)$$f_{\mathrm{cut}}\left(r_{ij}\right)=\left(1+\tanh\left(s_{ij}\left(r_{ij}-d_{ij}\right)\right)\right)/2$$
where *s_ij_* is a parameter that controls the strength of the damping for the interaction between atoms *i* and *j*, and *d_ij_* is the distance at which the cutoff function takes the value ½. As in our previous study, we set *s_ij_* = 10. The functional form of our corrections, i.e., Equation (1), was justified in our proof-of-concept work [21]. As shown in that paper, the differences between the reference and the PM6 IPECs have, in general, the form of typical intermolecular potential energy curves or the form of decaying exponentials with negative amplitudes, in agreement with Equation (1).

As already mentioned, the parameters were evaluated by fits to differences between the interaction energies calculated at the B3LYP-D3/def2-TZVP level and those computed with the PM6 method. The SQM calculations were carried out with the MOPAC2016 program [50]. We used a least-squares nonlinear fitting procedure based on a genetic algorithm [55,56], as implemented in our GAFit code [57], with the following objective function, *χ*^2^:(3)$$\chi^{2}\left(a\right)=\sum_{i=1}^{N}{\left[y_{i}-f(x_{i};a)\right]}^{2}\times w_{i}$$
where (*x_i_*, *y_i_*) represents one of the *N* data points, a is the collective variable formed by the total number of fitting parameters and *f* (*x_i_*; ***a***) is the value of the model function at *x_i_* (i.e., a particular geometry of the interacting molecules). The square of the difference between *y_i_* (i.e., a B3LYP-D3–PM6 energy difference) and the corresponding model value, calculated with Equations (1) and (2), may be multiplied by a weighting factor (*w_i_*), assigned to each data point. The use of a genetic algorithm has the advantage of efficiently exploring the search space and provides near optimal solutions when the number of fitting parameters is large. Since genetic algorithms may lead to many solutions that can be equally valid, our corrections should be viewed more as whole functional group corrections, rather than as individual pairwise corrections. 

## 4. Summary and Conclusions

Bearing in mind the key role noncovalent interactions play in chemistry and biology, we recently presented a new approach, i.e., the PM6-FGC method, to improve the performance of SQM methods in the description of noncovalent interactions [21]. The central idea is to develop pairwise analytical corrections, using a simple functional form that resembles Buckingham’s potential [58], but enhancing the parameter spectrum substantially. The parameters of the model function are adjusted by fits to differences between interaction energies calculated at the reference level and those evaluated with the SQM method (PM6 in our case) for complexes of small molecules, selected as representatives of functional groups. For each complex, we consider different orientations of the molecules, and include a series of configurations at several intermolecular distances. The inclusion of different orientations in the database is important to obtain well-balanced corrections. The corrections obtained should be considered as group corrections rather than as individual atom–pair corrections. To emphasize this, we chose the acronym FGC (Functional Group Corrections).

In our previous study [21], we used methane, formic acid, and ammonia as representatives to develop corrections for saturated hydrocarbons, carboxylic acids, amines, and, tentatively, amides. Although, in general, the PM6-FGC method provided quite good results, we found significant inaccuracies for some interactions involving –NH_2_ groups. For this reason, in the present study, we performed new parameterizations, using methylamine and acetamide as representatives of the amine and amide functional groups. The results of this work show a clear improvement over our previous parameterization and reinforce the importance of considering sufficient orientations of the interacting molecules in the reference database. We plan to extend the method to other functional groups relevant to biological compounds and implement the corrections in the MOPAC program. A Python script to calculate PM6-FGC corrections is available upon request.

## Figures and Tables

**Figure 1 molecules-27-01678-f001:**
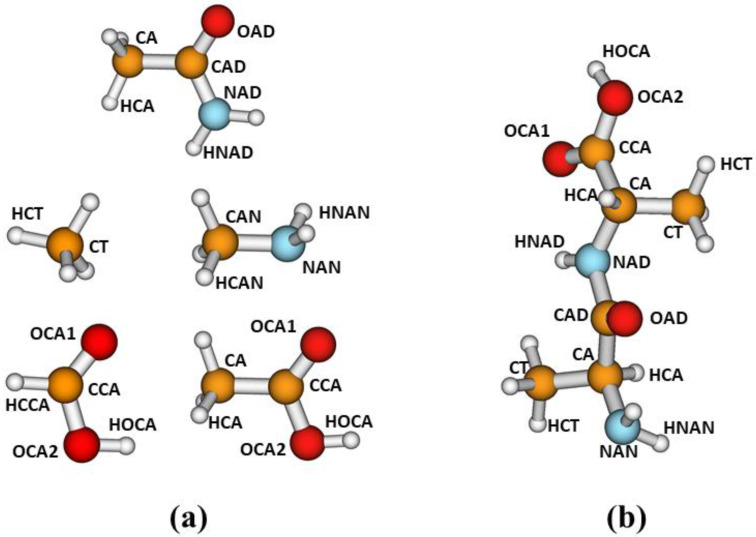
(**a**) Model compounds and atom types considered for the parameterizations, and (**b**) atom-type transferability to dialanine.

**Figure 2 molecules-27-01678-f002:**
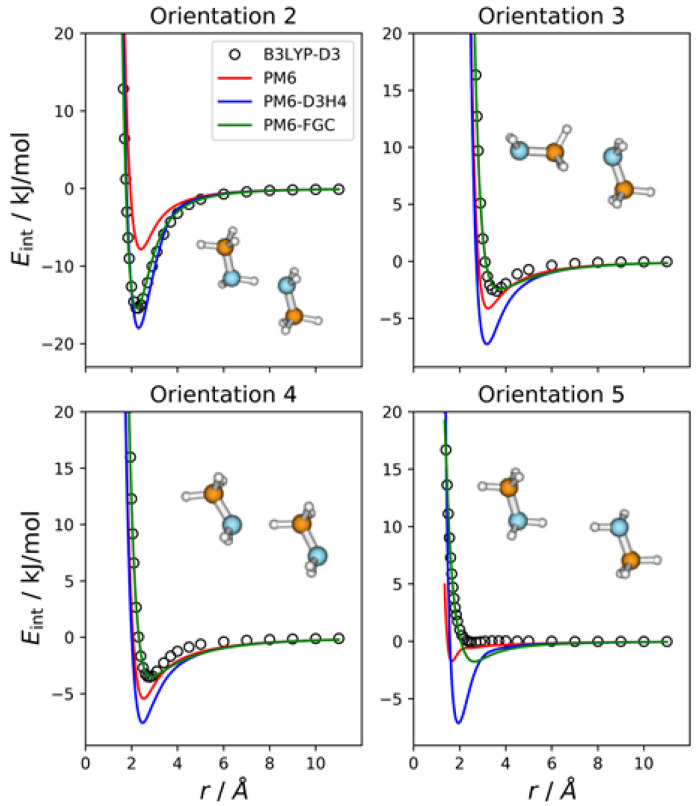
Comparison of IPECs for four selected orientations of the CH_3_NH_2_ dimer.

**Figure 3 molecules-27-01678-f003:**
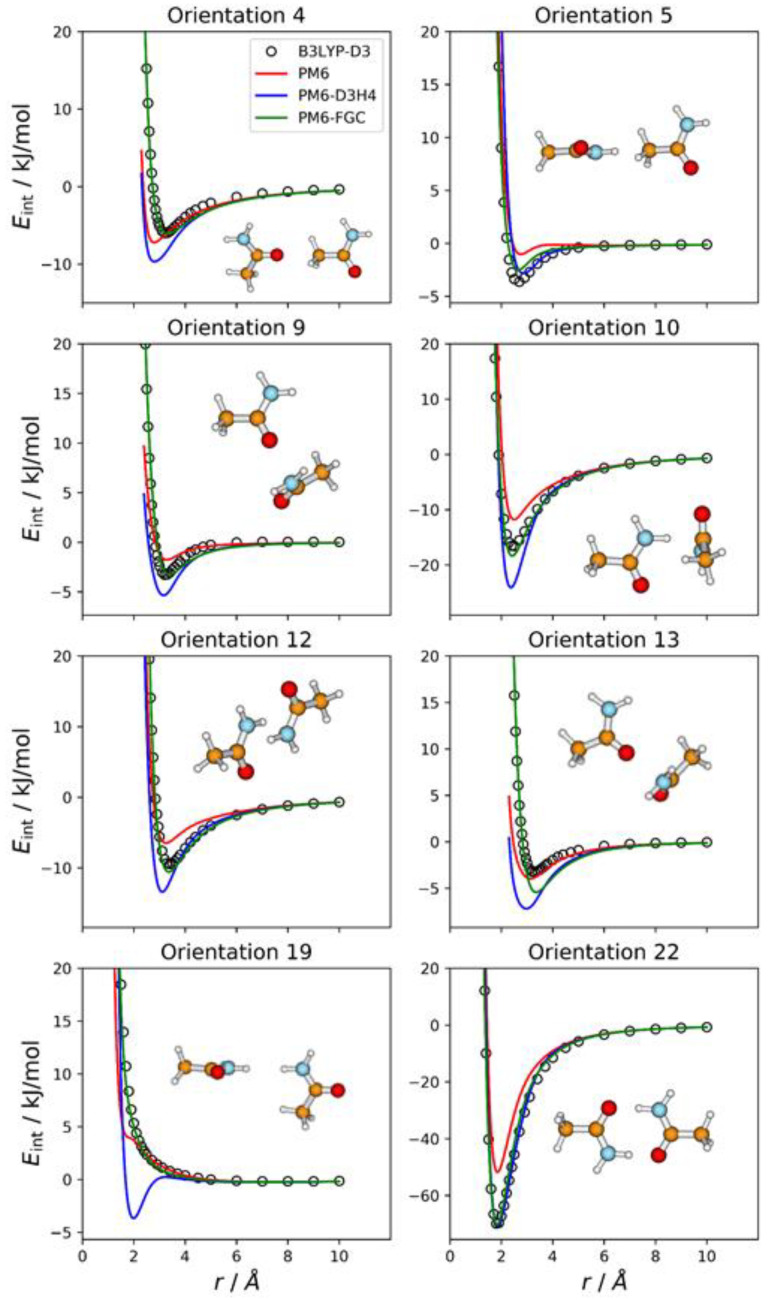
Comparison of IPECs for eight selected orientations of the CH_3_CONH_2_ dimer.

**Figure 4 molecules-27-01678-f004:**
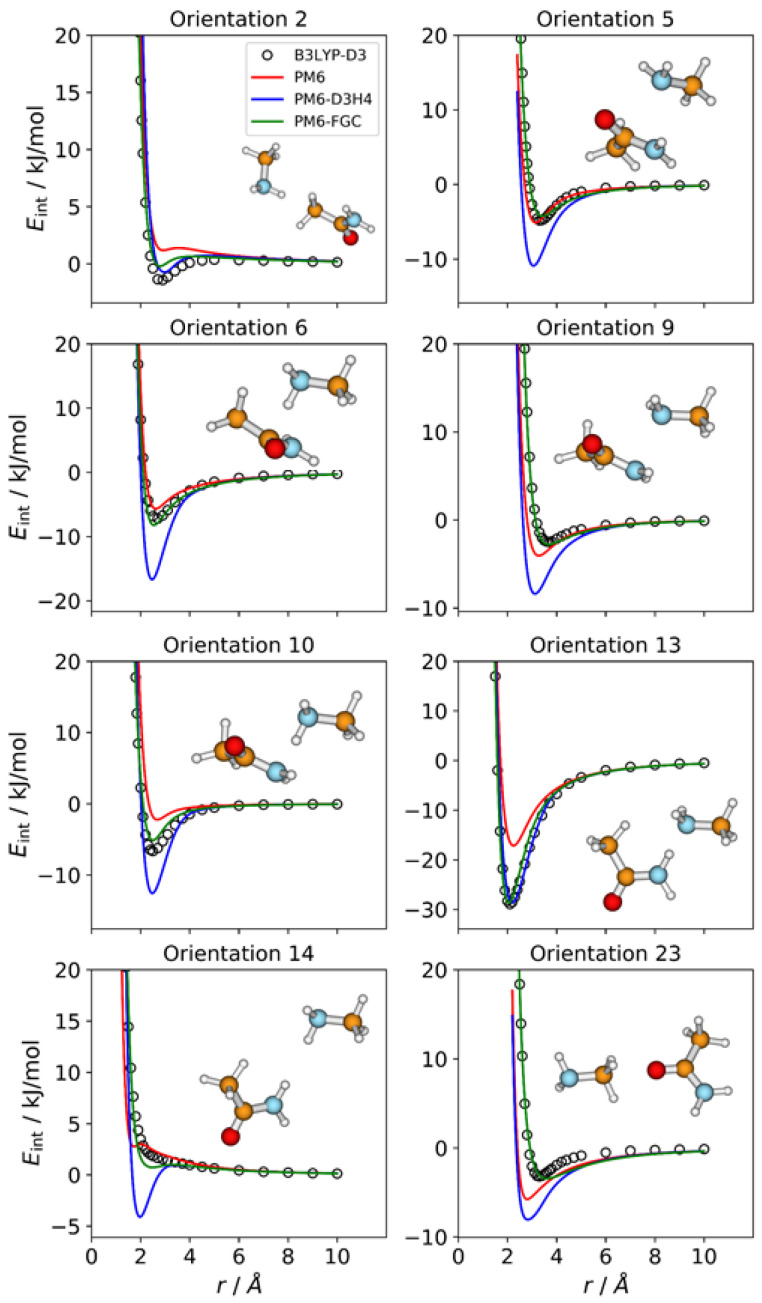
Comparison of IPECs for eight selected orientations of the CH_3_NH_2_–CH_3_CONH_2_ complex.

**Figure 5 molecules-27-01678-f005:**
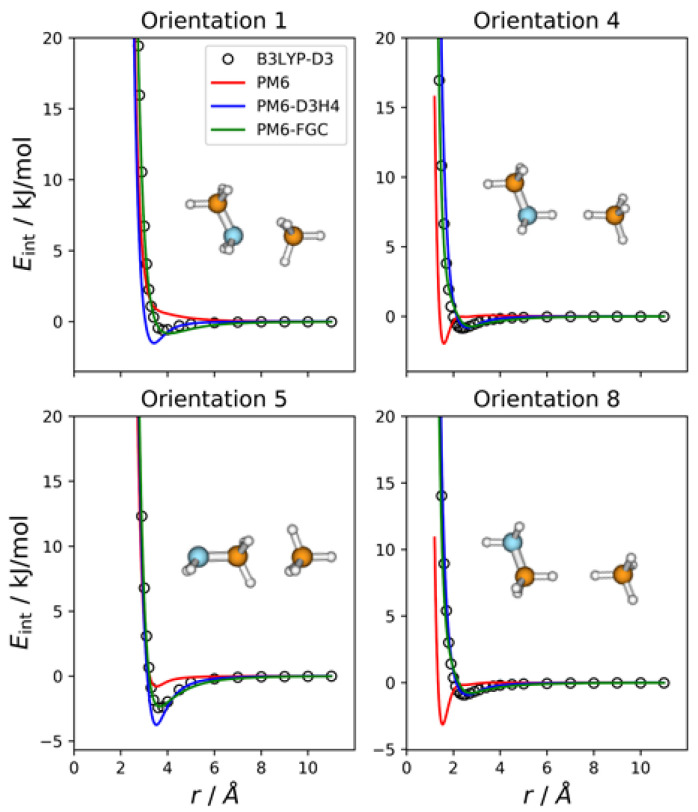
Comparison of IPECs for four selected orientations of the CH_3_NH_2_–CH_4_ complex.

**Figure 6 molecules-27-01678-f006:**
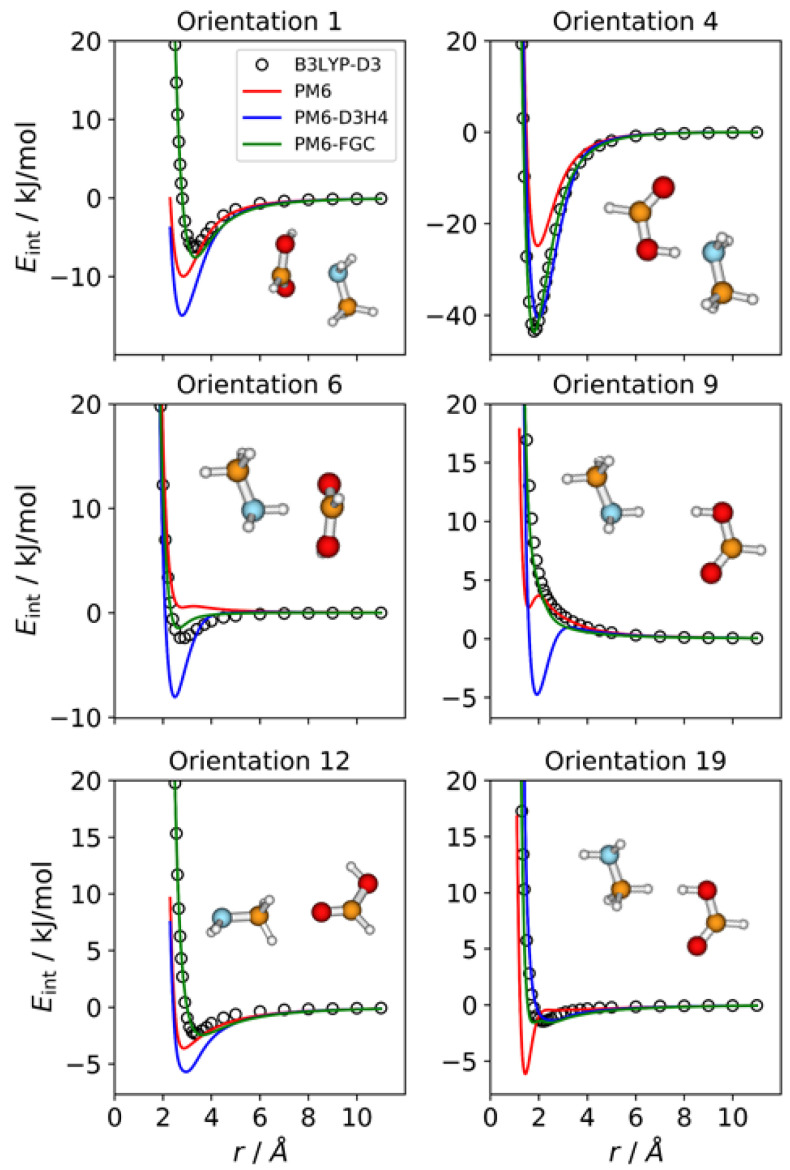
Comparison of IPECs for six selected orientations of the CH_3_NH_2_–HCOOH complex.

**Figure 7 molecules-27-01678-f007:**
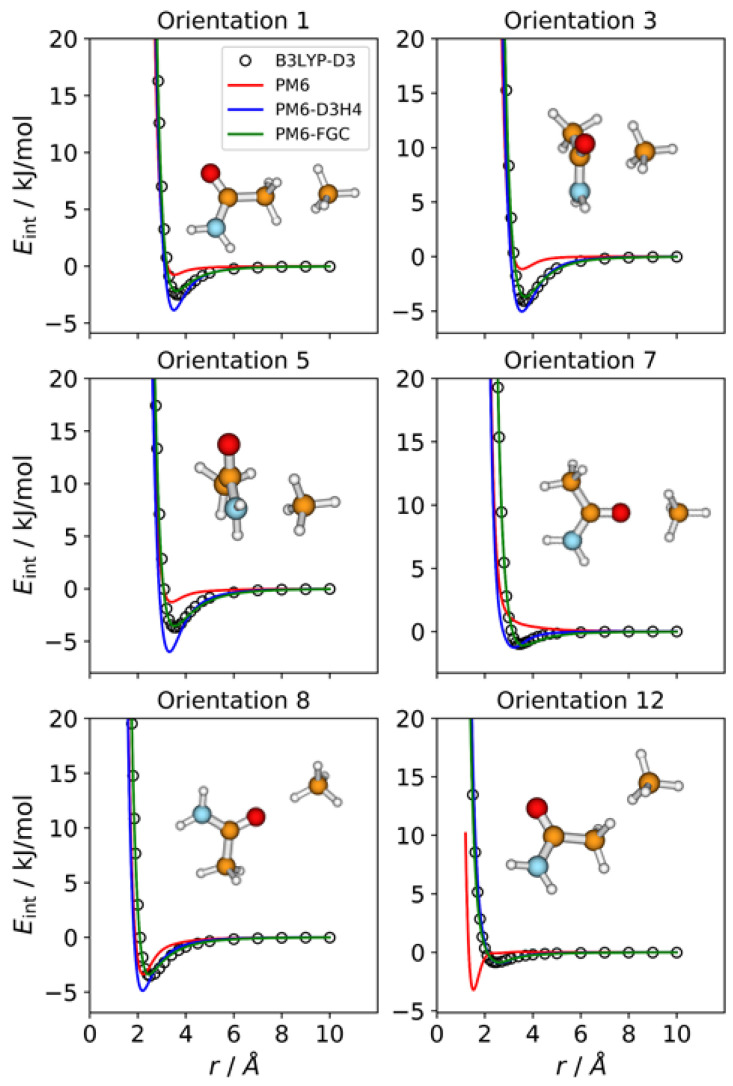
Comparison of IPECs for six selected orientations of the CH_3_CONH_2_/CH_4_ complex.

**Figure 8 molecules-27-01678-f008:**
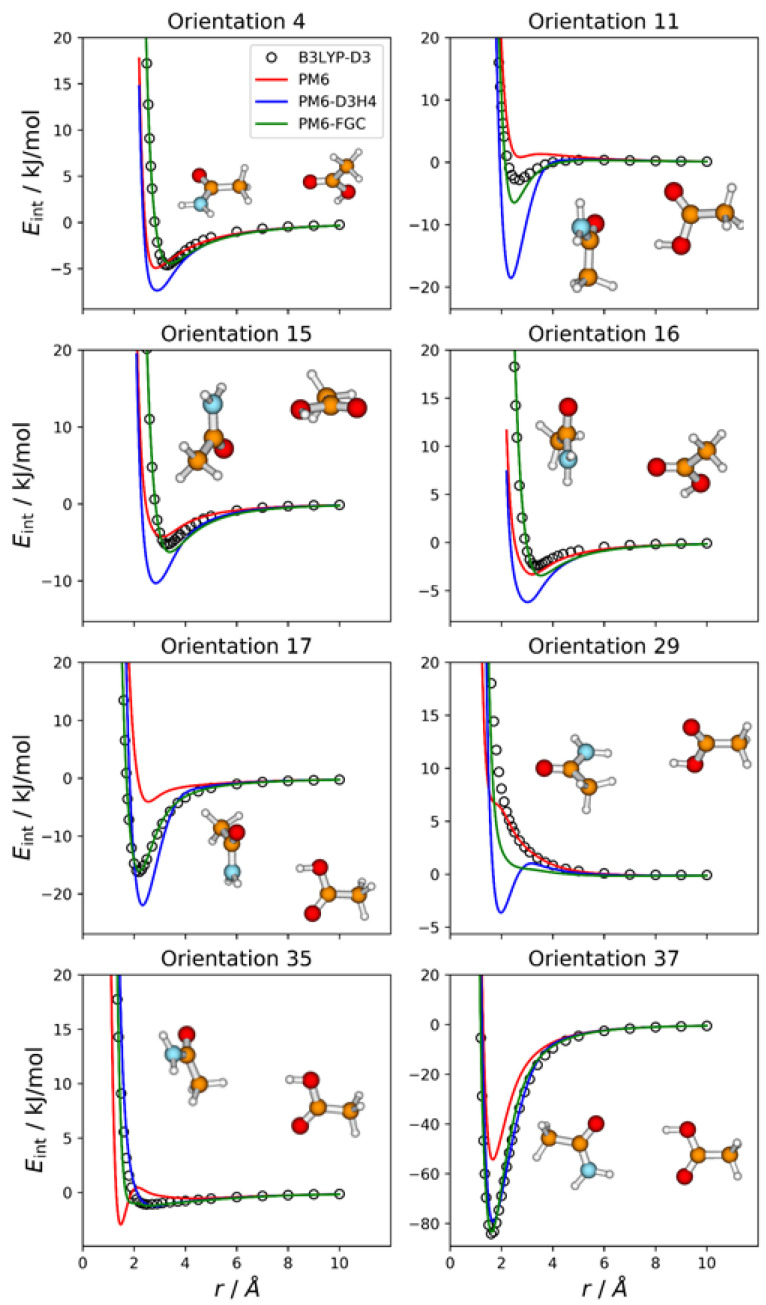
Comparison of IPECs for eight selected orientations of the CH_3_CONH_2_/CH_3_COOH complex.

**Figure 9 molecules-27-01678-f009:**
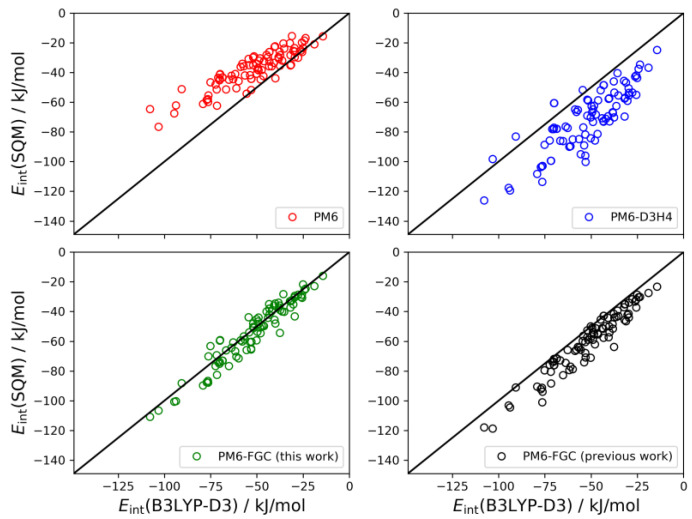
Linear correlations obtained for the dialanine dimer. For comparison, we include the PM6-FGC results evaluated with our previous corrections [21].

**Figure 10 molecules-27-01678-f010:**
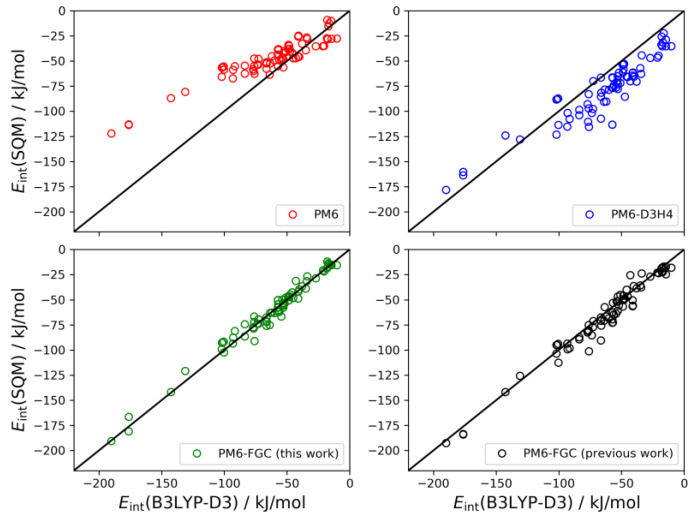
Linear correlations obtained for the diglycine dimer. For comparison, we include the PM6-FGC results evaluated with our previous corrections [21].

**Figure 11 molecules-27-01678-f011:**
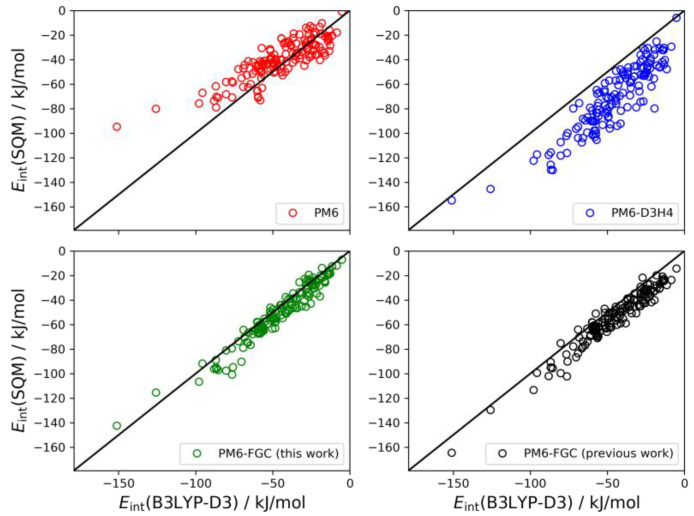
Linear correlations obtained for the diglycine trimer. For comparison, we include the PM6-FGC results evaluated with our previous corrections [21].

**Table 1 molecules-27-01678-t001:** Interaction energies (in kJ/mol) for selected complexes of the S66 database.

Complex ^a^	CCSD(T)/CBS ^b^	B3LYP-D3	PM6	PM6-D3H4	PM6-FGC ^c^	PM6-FGC ^d^
(**10**) CH_3_NH_2_–CH_3_NH_2_	−17.41	−18.04	−7.70	−19.00	−13.10	−14.10
(**11**) CH_3_NH_2_–peptide	−22.68	−22.71	−16.11	−25.69	−18.79	−19.02
(**14**) peptide–CH_3_NH_2_	−31.17	−32.98	−17.49	−31.42	−27.61	−32.32
(**15**) peptide–peptide	−36.11	−36.74	−24.73	−36.82	−30.12	−37.92
(**21**) CH_3_CONH_2_–CH_3_CONH_2_	−68.03	−70.09	−51.80	−70.71	−49.58	−70.95
(**46**) peptide–pentane	−17.82	−17.21	−5.27	−17.20	−17.57	−15.06
(**62**) pentane–CH_3_CONH_2_	−14.77	−14.46	−6.44	−16.48	−14.81	−13.47
MAE		0.87	11.21	1.51	5.21	2.42

^a^ Identification numbers used in the S66 database are given in parentheses. ^b^ Ref. [19]. ^c^ Previous work [21]. ^d^ This work.

**Table 2 molecules-27-01678-t002:** Statistical parameters ^a^ of the linear correlations using B3LYP-D3/def2-TZVP interaction energies as the reference.

	Diglycine Dimer		Dialanine Dimer		Diglycine Trimer	
	MAE	MBE	MAE	MBE	MAE	MBE
PM6	17.9	−15.0	14.3	−14.0	10.8	−5.1
PM6-D3H4	18.6	15.5	20.3	19.5	25.3	25.3
PM6-FGC	4.0 (6.0)	−0.6 (3.2)	4.9 (8.7)	1.9 (8.7)	5.9 (9.1)	4.6 (8.7)

^a^ MAE and MBE values are given in kJ/mol. Values in parentheses correspond to the parameterization performed in our previous work [21].

## Data Availability

Data are contained within the article and the Appendix A.

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
