# Peer review of "The PM6-FGC Method: Improved Corrections for Amines and Amides"

_molecules, 2022, doi:10.3390/molecules27051678_

Round 1
Reviewer 1 Report
see attachment

Reviewer 2 Report
This paper is of importance to modeling for in the area of drug design because of the amine and amide linkages present in amino acids and also in other biologically significant molecules. I applaud the authors for their thorough presentation and improvements to the PM6-FGC method. I have minor comments:
1) I will let the authors judge if some of the following references should be cited in their Introduction:
M. A. A. Ibrahim, J. Comput. Chem. 32, 2564 (2011)
M. Kolar and P. Hobza, J. Chem. Theory Comput. 8, 1325 (2012)
M. Carter et al, J. Chem. Theory Comput. 8, 2461 (2012)
W. L. Jorgensen and P. Schyman, J. Chem. Theory Comput. 8, 3895 (2012)
2) There are some recent papers that involve chalcogen bonding interactions that might be of interest. For example, see ChemPhysChem,19, 2540-2548 (2018).
Reviewer 3 Report
In a previous study (J. Chem. Theory Comput. 2021, 17, 5556-5567.), the authors developed the PM6 Functional Group Corrections (PM6-FGC) approach to correct the semiempirical PM6 Hamiltonian and improve the description of noncovalent interactions. Although, in general, the PM6-FGC method provided quite good results, the authors found significant inaccuracies for some interactions involving −NH2 groups. For this reason, in the present study they performed new parameterizations, using methylamine and acetamide as representatives of the amine and amide functional groups. The results of this work show a clear improvement over their previous parameterization and reinforce the importance of considering sufficient orientations of the interacting molecules in the reference database.
The main idea of this study has been published in JCTC. Someone maybe think that this work is only a correction to their previous study and the novelty of this work is very limited to publish it in a high impacted journal like Molecules. However, this study is indeed a progress in developing more reliable SQM methods. I have only one comment here:
The computations of the noncovalent interactions are always very challenging. The selection of suitable computational method is always case by case. I do not think that the B3LYP-D3/def2-TZVP results can be selected as the reference in this study. Considering that the dimers studied in this paper are very small, it should be not very expensive to perform the gold standard CCSD(T)/CBS calculations. I strongly suggest the authors to use the CCSD(T)/CBS values as the reference.
Author Response
Please, see the attachment.

Round 2
Reviewer 1 Report
Accept as it is.